# Peptide Antagonists for P-selectin Discriminate between Sulfatide-Dependent Platelet Aggregation and PSGL-1-Mediated Cell Adhesion

**DOI:** 10.3390/jcm8081266

**Published:** 2019-08-20

**Authors:** Suzanne J.A. Korporaal, Tom J.M. Molenaar, Bianca C.H. Lutters, Illiana Meurs, Sandra Drost-Verhoef, Johan Kuiper, Theo J.C. van Berkel, Erik A.L. Biessen

**Affiliations:** 1Division of Biopharmaceutics, Cluster BioTherapeutics, Leiden Academic Centre for Drug Research, 2300 Leiden, The Netherlands; 2Center for Circulatory Health, Department of Clinical Chemistry and Haematology, University Medical Center Utrecht, Utrecht University, 3584 Utrecht, The Netherlands; 3Laboratory of Experimental Cardiology, University Medical Center Utrecht, 3584 Utrecht, The Netherlands; 4Department of Pathology, University Medical Center Maastricht, 6229 Maastricht, The Netherlands; 5IMCAR, University Hospital RWTH Aachen, 52074 Aachen, Germany

**Keywords:** antagonists, peptides, platelets, P-selectin, sulfatide

## Abstract

Background: Membrane-exposed sulfatides are proposed to contribute to P-selectin-dependent platelet aggregation. Here, we demonstrated that P-selectin-mediated platelet aggregation on a collagen-coated surface under flow indeed depended on sulfatides and that this interaction differed considerably from the interaction of P-selectin with P-selectin Glycoprotein Ligand-1 (PSGL-1), which underlies leukocyte-endothelium adhesion. Methods and Results: Upon platelet activation, sulfatides were translocated to the platelet surface to form focal hot-spots. Interestingly, P-selectin was observed to exclusively interact with liposomes with a sulfatide density higher than 21% (*w*/*w*), indicating that the binding profile of P-selectin for sulfatide-rich liposomes was dependent on sulfatide density. Sulfatide-liposome binding to P-selectin and sulfatide/P-selectin-dependent platelet aggregation was blunted by peptide antagonists, carrying the EWVDV motif within N-terminal extensions, such as CDVEWVDVSC (half maximal inhibitory concentration IC_50_ = 0.2 μM), but not by the EWVDV core motif itself (IC_50_ > 1000 μM), albeit both being equally potent inhibitors of PSGL-1/P-selectin interaction (IC_50_= 7–12 μM). Conclusions: Our data suggest that the sulfatide/P-selectin interaction implicates multiple binding pockets, which only partly overlap with that of PSGL-1. These observations open ways to selectively interfere with sulfatide/P-selectin-dependent platelet aggregation without affecting PSGL-1-dependent cell adhesion.

## 1. Introduction

The selectin family of lectin-based adhesion molecules consists of L-selectin (in leukocytes), E-selectin (in endothelium), and P-selectin (in platelets and endothelium) [1,2]. The latter is stored in α-granules of platelets [3] and Weibel-Palade bodies of endothelial cells [4] and is rapidly translocated to the cell surface in response to a variety of inflammatory and thrombogenic stimuli [5,6]. Upon surface-exposure, P-selectin mediates platelet adhesion to leukocytes [7], mainly by interacting with its high-affinity ligand P-selectin glycoprotein ligand-1 (PSGL-1) [8,9]. Additionally, the P-selectin/PSGL-1 interaction induces platelet activation, thereby enhancing platelet aggregation and thrombus formation [10]. Interestingly, P-selectin has also been found to interact with sulfatides (3-sulfated galactosyl ceramides) [11,12], which are glycosphingolipids of the platelet cell membrane [13]. It should be noted that sulfatides have been reported to bind laminin [14], thrombospondin [14], von Willebrand Factor [14], midkine [15], annexin V [16], L-selectin [11,17], and possibly E-selectin as well [18,19].

Merten and colleagues were the first to show that P-selectin binds to sulfatides exposed on the platelet surface upon activation, and that platelet aggregates are stabilized through sulfatide/P-selectin rather than PSGL-1/P-selectin interactions. By inducing further degranulation, thereby increasing surface expression of P-selectin, P-selectin/sulfatide interactions trigger a positive feedback mechanism that potentiates aggregation [20,21,22,23]. Interference of P-selectin binding to sulfatides results in attenuated stability of platelet aggregates in vitro and ex vivo [20,21,22,23]. Accordingly, the P-selectin/sulfatide interaction is tightly controlled by Disabled-2 (Dab2), which reduces P-selectin expression levels on platelets and aggregate size after binding of Dab2 to sulfatides [24]. The results from these studies may well explain the slight bleeding tendency observed in P-selectin-deficient mice [25]. Hence, the development of antagonists of the sulfatide/P-selectin interaction may benefit in the treatment of various pathophysiological conditions, such as acute coronary symptoms, which is underlined by the observation that sulfatides are attributed a role in P-selectin dependent neointima formation [26] and breast cancer progression [27]. However, the molecular mechanism underlying the P-selectin/sulfatide interaction and its implications for P-selectin-directed intervention in thrombosis remains poorly understood.

We previously reported the design of specific peptide antagonists for P-selectin [28,29] and showed that peptides carrying an EWVDV minimal motif interacted with P-selectin close to the PSGL-1 binding site on P-selectin, which was proposed to overlap with that of sulfatides [30,31].

In the present study, we exploited this feature to further characterize sulfatide binding to P-selectin using a panel of P-selectin blocking peptides and to investigate the ability of the peptide antagonists to prevent sulfatide binding to P-selectin, and P-selectin-mediated platelet aggregation. We demonstrated that P-selectin binding to sulfatide-laden liposomes required a threshold sulfatide density, that activated platelets express clusters of surface-exposed sulfatides, and that the binding profile of P-selectin to sulfatides differed from that to PSGL-1. Finally, we showed that elongated but not truncated EWVDV peptide antagonists inhibited sulfatide binding to P-selectin, as well as platelet aggregation, which paves the way to the development of new strategies for selective therapeutic intervention in thrombus size and stability.

## 2. Materials and Methods

### 2.1. Materials

We obtained human P- and E-selectin-IgG from R&D systems (Minneapolis, MN, USA), human L-selectin-IgG and monoclonal anti-P-selectin antibodies AK-4 and AC1.2 from Pharmingen (San Diego, CA, USA), adenosine diphosphate (ADP), sulfatides (cerebroside sulfate, bovine), cholesterol, goat-anti-human IgG (Fc-specific), human placental collagen type III and filipin III from Sigma (St. Louis, MO, USA), fibrillar equine collagen from Hormon Chemie (München, Germany), prostacyclin (PGI_2_) from Cayman Chemical (Ann Arbor, MI), human fibrinogen from Kordia Life Sciences (Leiden, The Netherlands), human serum albumin from ICN Biomedicals Inc. (Aurora, OH, USA), Alexa Fluor^®^ 488 goat-anti-mouse IgG from Molecular Probes (Leiden, The Netherlands), Aqua/Poly Mount from Polysciences Inc. (Warrington, PA, USA), egg yolk phosphatidylcholine from Lipoid (Ludwigshafen, Germany), and [^3^H]-cholesteryloleate from Amersham Life Sciences (Little Chalfont, UK). Peptides were purchased from Eurosequence (Groningen, The Netherlands), and their quality was confirmed by mass spectroscopy and HPLC. Antibody WAPS12.2 was isolated from HB-299 cells obtained from the American Type Culture Collection (ATCC).

### 2.2. Mice

Six to eight week old male wild type C57Bl6 mice (referred to as wildtype (WT)) and P-selectin-deficient mice (P-selectin knockout (KO)) on a C57Bl6 background were obtained from Jackson Laboratories (Bar Harbor, Maine). Mice had unlimited access to water and regular chow diet (RM3, Special Diet Services, Witham, UK). Blood was obtained from surplus mice scheduled for sacrifice at the Leiden animal breeding facility. Animal experiments were performed at the Gorlaeus Laboratories of the Leiden Academic Center for Drug Research following the national laws and approved by the local ethics committee for animal experiments.

### 2.3. Preparation of Sulfatide Containing Liposomes

Liposomes were prepared by sonication as follows. Egg yolk phosphatidylcholine, cholesterol, and sulfatides, all dissolved in chloroform/methanol (1:1, v/v), were mixed at weight ratios of 4:0.8:0 (named Sf0), 4:0.8:1.3 (Sf21), 2.5:0.8:1.3 (Sf28), 1:0.8:1.3 (Sf42), and 0.2:0.8:1.3 mg/mL (Sf57). The Sf numbers indicate the number of sulfatides as a percentage of total lipid weight. A tracer amount of [^3^H]-cholesteryl oleate ([^3^H]-CE; 1.3 × 10^8^ dpm (disintegrations per minute)) was added, and the organic solvent was evaporated by a stream of nitrogen. The resulting lipid layer was vortexed in 1 mL phosphate buffered saline (PBS) and subsequently sonicated for 40 min at 54 °C using a Soniprep 150 (MSE Scientific Instruments, Crawley, UK) at 18 µm amplitude. Particle sizes were determined by photon correlation spectroscopy (Malvern 4700 C System, Malvern Instruments, Malvern, UK) at 25 °C and a 90° angle between laser and detector. Sonication resulted in monodisperse liposomes with mean particle diameters between 63–85 nm (polydispersities of 0.13–0.22). Liposomes were stored at 4 °C under argon and used for characterization and metabolic studies within 7 days following preparation, during which no physicochemical changes occurred. As the liposomes were equal in size and incorporation of the radioactive label was quantitative, the specific activity of each sulfatide-bearing liposome was calculated as the amount of [^3^H]-cholesteryloleate (dpm) divided by the total lipid content (ng).

### 2.4. Liposome Interaction with Selectins

Fc-specific goat anti-human IgG (10 μg/mL) was coated overnight at 4 °C in a high binding 96-well plate (Costar, Corning, UK) in coating buffer (50 mM NaHCO_3_, pH 9.6). Then, wells were washed with assay buffer (20 mmol/L HEPES, 150 mmol/L NaCl, 1 mmol/L CaCl_2_, pH 7.4) and incubated for 1 h at 37 °C with blocking buffer (3% bovine serum albumin (BSA) in assay buffer). After washing, wells were incubated with fusion proteins of IgG and P-, E-, or L-selectin (0.3 μg/mL, 2 h, 37 °C), washed with assay buffer, and subsequently incubated for 2 h at 4 °C with a concentration range of [^3^H]-CE-labeled liposomes (0–800 µg lipid per mL assay buffer containing 0.2% BSA). For competition studies, labeled liposomes (1 × 10^5^ dpm/well) were incubated in the presence of peptides or antibodies at the indicated concentrations. After removal of unbound liposomes by washing four times with assay buffer, bound liposomes were collected in 2-propanol/butanol/water (50:25:20, by vol.) and counted for radioactivity using a scintillation counter. Since the sulfatide liposomes were equal in size, the recovered amount of [^3^H]-CE could be used to calculate the binding capacity of P-selectin for the sulfatide liposomes, as well as the recovered amount of lipid (ng/well) based on the specific activity.

### 2.5. Surface Plasmon Resonance (Biacore) Analysis

A Biacore 2000 instrument (Biacore AB, Uppsala, Sweden) was used to analyze the interaction between sulfatides and P-, L, and E-selectins, as described [28]. Experiments were performed at room temperature using running buffer (20 mmol/L HEPES, 150 mmol/L NaCl, 1 mmol/L CaCl_2_, 0.005% Surfactant P20, pH 7.4). Sulfatides were immobilized on L1 sensor chips at 5 μL/min according to the manufacturer’s instructions. Binding kinetics of human fusion proteins IgG and P-, L- and E- selectins to the immobilized sulfatides were determined by injection of the fusion proteins. At the end of each injection, the sensor chips were regenerated with 100 mmol/L NaOH and equilibrated with running buffer. All curves were corrected for non-specific binding by an online baseline subtraction of ligand binding to a control flow channel. Binding kinetics were analyzed using BIAevaluation software (V2.1; Pharmacia Biosensor, Uppsala, Sweden).

### 2.6. Blood Collection and Platelet Isolation

Freshly drawn venous blood from healthy volunteers was obtained through the Mini Donor Service, a blood donation facility for research purposes, that is approved by the Medical Ethics Committee of the University Medical Center Utrecht (protocol number 07-125). All donors provided written informed consent, in accordance with the Declaration of Helsinki for blood collection through this service. All participant information is kept confidential, and only Mini Donor Service personnel have access to the information provided. The donors were free of medication for at least two weeks before blood collection. Overnight fasting before blood collection was not necessary. Blood was collected into 0.1 volume 130 mmol/L trisodium citrate and was processed the same day and within 6 h. For aggregation studies, platelet-rich plasma (PRP) was prepared by centrifugation (150 g, 15 min, 20 °C) and adjusted to a final concentration of 2×10^11^ platelets/L with platelet-poor plasma prepared from the remaining blood (1100 g, 10 min, 20 °C). Washed platelets were prepared by adding 0.1 volume of anticoagulant citrate dextrose (ACD) solution (2.5 g trisodium citrate, 1.5 g citric acid, and 2 g D-glucose in 100 mL distilled water) to lower the pH of the PRP to 6.5, followed by two cycles of centrifugation (330 g, 15 min, 20 °C) and resuspension in HEPES-Tyrode buffer (145 mmol/L NaCl, 5 mmol/L KCl, 0.5 mmol/L Na_2_HPO_4_, 1 mmol/L MgSO_4_, 10 mmol/L HEPES, 5 mmol/L D-glucose, pH 6.5). Prostacyclin (PGI_2_, final concentration (f.c.) 10 ng/mL) was present during the final wash step, and the final resuspension was in HEPES-Tyrode buffer (pH 7.2) to a concentration of 2 × 10^11^ platelets/L. Experiments were performed with blood or platelets from at least 3 donors.

For studies with murine platelets, six to eight-week-old male WT and P-selectin KO mice were anesthetized by subcutaneous injection of a mixture of xylazine (5 mg/mL), ketamine (0.04 mg/mL), and atropine (0.05 mg/mL) and exsanguinated by extraction via an intracardiac syringe. Blood was anticoagulated with 0.1 volume 130 mmol/L trisodium citrate, recalcified, and used for perfusion studies. For static adhesion experiments, washed mouse platelets were prepared by centrifugation of whole blood (300 g, 3 min, 20 °C). Plasma and buffy coat were isolated, and PRP was concentrated by centrifugation (700 g, 15 sec, 20 °C), followed by another centrifugation step (2000 g, 2 min, 20 °C) in the presence of ACD and PGI_2_. Pellets were resuspended in HEPES-Tyrode buffer (pH 6.5), and the washing procedure was repeated once. The platelet pellets were subsequently resuspended in HEPES-Tyrode buffer (pH 7.2), and the platelet count was adjusted to 2 × 10^11^ platelets/L.

### 2.7. Static Adhesion

Washed mouse platelets were stimulated with vehicle or thrombin (1 U/mL, 1 min, 20 °C) and coated on a high-binding 96-well plate (1 h, 37 °C). Then, wells were washed with assay buffer (20 mmol/L HEPES, 150 mmol/L NaCl, 1 mmol/L CaCl_2_, pH 7.4) and incubated with blocking buffer (3% BSA in assay buffer, 1 h, 37 °C). After washing, wells were incubated (2 h, 37 °C) with [^3^H]-CE-labeled liposomes (Sf28, 1 × 10^5^ dpm/well) in the absence or presence of the monoclonal anti-sulfatide antibody SulphI (5 μg/mL). After removal of unbound liposomes by washing four times with assay buffer, bound liposomes were collected in 2-propanol/butanol/water (50:25:20, by vol.) and counted for radioactivity using a scintillation counter.

### 2.8. Perfusion Studies

Adhesion of human and mice platelets under flow was studied in a single passage perfusion chamber under nonpulsatile flow using a modified parallel plate perfusion chamber with a slit width of 2 mm and a slit height of 0.1 mm. Human fibrinogen was coated on glass coverslips (18 × 18 mm; Menzel Gläser, Braunschweig, Germany) at a concentration of 100 μg/mL for 1 h at room temperature. Human placental collagen type III (solubilized overnight in 50 mmol/L acetic acid) or fibrillar equine collagen (1 mg/mL) were sprayed onto Thermonox coverslips at a density of 30 μg/mL with a retouching airbrush (Badger model 100 LGF, Badger Brush Co, Franklin Park, IL, USA) under a nitrogen-operating pressure of 1 atm. Coverslips were blocked with human albumin (1% in PBS) and stored overnight at 4 °C.

Prewarmed human and murine blood samples (5 mL and 1 mL, respectively) were drawn through the perfusion chamber by an infusion pump (pump 22, model 2400-004; Harvard, Natick, MA, USA) for 3 (human) and 5 min (mouse). Adhesion of human platelets to fibrinogen was performed at a shear rate of 300 s^−1^ (representative for large arteries) and to collagen type III at 800 s^−1^ (representative for small arteries); adhesion of murine platelets to fibrillar equine collagen was at 150 and 300 s^−1^ in the absence and presence of Sulph I antibody (10 µg/mL), respectively. Higher shear rates required large volumes of murine blood, which was unfortunately not available.

Coverslips with human platelets were rinsed with HEPES-buffered saline (10 mmol/L HEPES, 150 mmol/L NaCl, pH 7.4) and fixed with 1% paraformaldehyde in PBS for immunofluorescence studies. For evaluation by light microscopy, coverslips with murine platelets were fixed with 0.5% glutaraldehyde in PBS, stained with May-Grünwald-Giemsa, dried and mounted in Entellan (Merck, Darmstadt, Germany), and analyzed with a light microscope (Leitz, Wetzlar, Germany) equipped with a JAI-CCD camera (Copenhagen, Denmark) coupled to a Matrox frame grabber (Matrox Electronic Systems Ltd., Dorval, Quebec, Canada) using Optimas 6.2 software (Optimas Inc., Seattle, WA, USA). Murine platelet deposition was expressed as percentage surface coverage assessing 20 fields in a randomized manner.

### 2.9. Immunofluorescence Studies

Coverslips from perfusion experiments with human blood were washed with PBS, incubated in 50 mmol/L NH_4_Cl/PBS (10 min, 20 °C), washed again and blocked with PBS containing 3% BSA (10 min, 37 °C). To detect membrane cholesterol, coverslips were incubated with 50 μg/mL filipin III in PBS/1% BSA (1 h, 20 °C). For detection of sulfatides, coverslips were incubated with 1 μg/mL SulphI in PBS/3% BSA (1.5 h, 20 °C), washed and incubated with Alexa Fluor^®^ 488 goat-anti-mouse IgG (1:1000 diluted in PBS/3% BSA, 1 h, 20 °C). Finally, coverslips were washed with PBS and mounted in Aqua/Poly Mount. Cells were visualized with a Nikon Eclipse E600 fluorescence microscope (60x Nikon objective) equipped with a CoolSNAP-Pro camera (Media Cybernetics, Inc., Silver Spring, MD, USA), using Image-Pro Plus software (Media Cybernetics, Inc., Rockville, MD, USA) for analysis.

As controls, resting platelets were spun down onto microscope slides (500 rpm, 5 min, 20 °C), fixed with 1% paraformaldehyde, and immunofluorescently labeled as described above.

### 2.10. Analysis of Platelet Aggregation

Human PRP was incubated with peptide antagonists (100 μmol/L, 5 min, 37 °C) or vehicle and stimulated with 10 μmol/L ADP under constant stirring at 1000 rpm (37 °C). Optical aggregation was monitored in a Chrono-Log Lumi-aggregometer (Chrono-Log Corporation, Havertown, PA, USA).

After 15 min, aggregates were fixed in 1 volume 0.5% glutaraldehyde in PBS and applied to microscope slides. Aggregates were dehydrated with methanol and stained with May-Grünwald/Giemsa. The aggregate size was measured by light microscopy equipped with a JAI-CCD camera (Copenhagen, Denmark) coupled to a Matrox frame grabber (Matrox Electronic Systems Ltd., Dorval, Quebec, Canada) using the ‘watershed’ function of the Optimas 6.2 software (Optimas Inc., Seattle, WA, USA). Results were expressed as mean aggregate size (μm^2^).

### 2.11. Statistical Analysis

Data were expressed as mean ± S.E.M. with n number of observations and were analyzed by the Student’s t-test for unpaired observations. Differences were considered significant at *p* < 0.05.

## 3. Results

### 3.1. Sulfatide P-selectin Tethering

Merten et al. have previously shown that sulfatide/P-selectin interactions contribute to human platelet aggregate stability [21,22]. In an attempt to better understand the actual role of sulfatides in platelet-platelet interactions, the binding of sulfatide-laden liposomes to mouse platelets was investigated under static conditions. Activated platelets from WT and P-selectin KO mice were stably coated to wells, and after blocking, incubated with 28% sulfatide-laden [^3^H]-CE labeled liposomes. Liposome binding to WT platelets in the absence or presence of the sulfatide binding blocking antibody SulphI did not differ and was in the same range as binding to P-selectin KO platelets. Thus, under static conditions, sulfatides did not bind to P-selectin expressed on adhered platelets.

To assess the role of sulfatides under flow, murine blood was perfused over a surface coated with equine collagen at a shear rate of 150 s^−1^, and the platelet adhesion was measured. The percentage surface coverage of P-selectin KO platelets (23.1 ± 4.4%) was 33% lower than that of WT platelets (34.6 ± 3.9%; *p* = 0.028). Importantly, adhesion of WT platelets was reduced by 41% by the SulphI antibody to 20.3 ± 1.8% (*p* = 0.0047), but with P-selectin KO platelets, the antibody had no effect (18.4 ± 2%; *p* = 0.17) (Figure 1A). Essentially, similar results were found at a higher shear rate of 300 s^−1^: the anti-sulfatide antibody SulphI reduced adhesion of WT platelets by 50%, whereas no inhibition (−7%; *p* = 0.47) was observed for P-selectin KO platelets. In line with earlier observations [22], SulphI antibody interfered with the aggregate density suggestive of weakened platelet-platelet interaction (Figure 1B,C). Thus, the interaction between sulfatides and P-selectin contributed to platelet adhesion and aggregate formation to collagen under flow, but not under static conditions.

We next investigated whether the cellular distribution of sulfatides was affected upon platelet activation, in analogy to previous observations regarding platelet cholesterol [32]. Hereto, human platelets were perfused over immobilized fibrinogen and collagen type III. Filipin III staining for cholesterol showed only low levels of exposed cholesterol in resting platelets (Figure 2A), whereas platelets adhered to fibrinogen (Figure 2C) and collagen (Figure 2E) revealed cholesterol-rich foci at the plasma membrane. In fibrinogen-adhered platelets, cholesterol mainly accumulated at tips of filopodia. Immunofluorescent labeling with the anti-sulfatide antibody SulphI [33] showed a similar pattern with faint surface staining of resting platelets (Figure 2B), but avid and focal staining of platelets adhered to fibrinogen (Figure 2D) and collagen (Figure 2F). As expected, strong activation of platelets by collagen led to aggregate formation, accompanied by an even further increase in surface-exposed sulfatides. Thus, platelet adhesion under flow triggered surface expression of both cholesterol and sulfatide clusters.

### 3.2. Sulfatide Surface Density is Important for Interaction with P-selectin

To investigate whether the increase in sulfatide density seen on activated platelets contributes to P-selectin binding, liposomes were prepared containing 0, 21, 28, 42, and 57% sulfatides (defined as Sf0, Sf21, Sf28, Sf42, and Sf57), relative to total lipid content (*w*/*w*), and the interaction with P-selectin was measured in an ELISA setup (Figure 3A). P-selectin binding of sulfatide liposomes depended on the sulfatide density, with high-density liposomes binding more avidly than their low-density counterparts. Analysis of the liposome binding kinetics revealed a progressive increase in affinity for P-selectin with increasing sulfatide density, resulting in dissociation constants (Kd’s) of 450 ± 40, 81 ± 6, and 66 ± 10 μg/mL (equivalent to 57 ± 5, 10.3 ± 0.7, and 8.4 ± 1.3 nM, n = 3) for Sf28, Sf42, and Sf57 liposomes, respectively. P-selectin binding of liposomes containing 21% sulfatides was barely detectable, and control liposomes (Sf0) displayed only background binding. Thus, effective binding of sulfatide liposomes to P-selectin required an exposed sulfatide density higher than 21%.

To determine the specificity of the interaction of sulfatide-containing liposomes with P-selectin, we investigated the binding of Sf21, Sf28, Sf42, and Sf57 liposomes to P-, E-, and L-selectin. Both P- and L-selectin, but not E-selectin, bound the sulfatide liposomes with high affinity (Figure 3B). A similar preference for P- and L-selectin binding was observed by plasmon surface resonance spectroscopy using sulfatide coated chips (Figure 3C). As for P-selectin, the interaction of sulfatide liposomes with L-selectin depended strongly on sulfatide density on the liposomal surface, and also here a threshold liposomal sulfatide content (21%) was required for adequate interaction. This could point to multitethered interactions between sulfatides and P- or L-selectin.

### 3.3. Characterization of the Interaction of Sulfatide Liposomes with P-selectin

The notion that multiple binding sites contribute to the interaction of sulfatide liposomes with P-selectin was confirmed in subsequent binding studies with a panel of (selective) P-selectin inhibitors. Monoclonal anti-P-selectin blocking antibodies AK-4 and WAPS12.2 (10 µg/mL), rPSGL-1-Ig (15 µg/mL), the Ca^2+^ chelator EDTA (10 mM), which prevents the binding of carbohydrate to P-selectin, and the peptide antagonist CDVEWVDVSC (0.5 µM), known to block P-selectin binding to PSGL-1 [28], all blunted the binding of Sf28 liposomes to P-selectin (−84%; *p* < 0.0001, −97%; *p* < 0.0001, −88%; *p* = 0.0061, −79%; *p* = 0.0068, and −63%; *p* = 0.03, respectively) (Figure 3D). Surprisingly, blocking antibody AK-4 only marginally affected binding of Sf42 liposomes (−36%; *p* = 0.0032) and completely failed to inhibit Sf57 liposome binding. WAPS12.2 and rPSGL-1-Ig interfered strongly with Sf42 binding (−99%; *p* < 0.0001 and −97%; *p* = 0.0003, respectively). For Sf57 liposome binding, complete (WAPS12.2, −99%; *p* < 0.0001) and partial (rPSGL-1-Ig, −68%; *p* < 0.0001) inhibition was found. EDTA and the peptide antagonist did not affect the binding of Sf42 and Sf57 liposomes to P-selectin. Non-blocking monoclonal antibody AC1.2 failed to change liposome binding. These findings establish that the binding of sulfatide-rich liposomes involves the carbohydrate recognition domain of P-selectin and that binding sites for sulfatides and carbohydrates do not completely overlap.

EWVDV-based peptide P-selectin antagonists were used to further map the interaction of sulfatides with P-selectin. Displacement studies of the binding of [^3^H]-CE-labeled sulfatide liposomes to P-selectin were performed for a selection of peptide antagonists, derived from the parent peptide TM11 (CDVEWVDVSSLEWDLPC) [28]. Sf28 liposome binding to P-selectin was dose-dependently inhibited by the elongated EWVDV peptides DVEWVDVS (EC_50_ = 12 μM), DVEWVDV**A** (EC_50_ = 8 μM), and CDVEWVDVSC (EC_50_ = 0.2 μM). Surprisingly, the truncated peptides EWVDV, VEWVDV, **A**VEWVDVS, D**A**EWVDVS, and DV**A**WVDVS, all potent inhibitors of adhesion of HL60 cells to P-selectin [28], did not change sulfatide liposome binding to P-selectin at all. The inactive control peptides DVE**A**VDVS, DVEW**A**DVS, DVEWV**A**VS, DVEWVD**A**S, and EWV**K**V [28] did not alter the binding of sulfatide liposomes to P-selectin either (Figure 3E and Table 1). The minimal motif necessary to block the interaction of P-selectin with PSGL-1 was EWVDV (ref. [28] and Table 1), whereas inhibition of sulfatide/P-selectin interaction required N-terminal elongation of the peptide sequence.

### 3.4. Peptide Antagonists Inhibit Platelet Aggregation

Since the interaction of sulfatides with P-selectin is instrumental in platelet aggregation (ref. [22] and Figure 1), we investigated whether aggregation in stirred platelet suspensions was sensitive to the peptide antagonists. Human platelets were activated by ADP, and platelet aggregation was measured in the absence or presence of EWV**K**V, EWVDV, DVEWVDVS, and CDVEWVDVSC. Peptide antagonists DVEWVDVS and CDVEWVDVSC significantly reduced maximum aggregation by 28% (*p* < 0.05) and 43% (*p* < 0.001), respectively (Figure 4 A,B); the initial slope of platelet aggregation remained unaltered. In keeping with the inability of EWVDV to displace sulfatide liposome binding to P-selectin, EWVDV failed to impact aggregation, although this peptide was previously shown to be an equally potent inhibitor of P-selectin/PSGL-1 binding as the full-length antagonist [28]. Anti-P-selectin antibody AK-4 (10 μg/mL) and anti-sulfatide antibody SulphI (10 μg/mL) inhibited platelet aggregation by 90% (*p* < 0.0001) and 20% (*p* < 0.05), respectively, confirming a role for sulfatide/P-selectin interactions in platelet aggregation [22].

Both the peptide antagonist CDVEWVDVSC and the antibodies AK-4 and SulphI reduced the mean aggregate size by about 90% (Figure 5A), in line with earlier findings [21]. While in the absence of inhibitors, only 15% of aggregates were smaller than 1000 μm^2^, this number increased to 76%, 85%, and 83% in the presence of CDVEWVDVSC, SulphI, and antibody AK-4, respectively (*p* < 0.0001) (Figure 5B–E). These intriguing results indicated that despite the partial inhibitory effect of CDVEWVDVSC and SulphI on maximal platelet aggregation, both had profound effects on aggregate size.

## 4. Discussion

Various negatively charged polymers, including heparin [34], sulfated glycans [35], and sulfatides [11,12] have been reported to bind P-selectin, and, in particular, sulfatides are regarded to be prime counterligands in P-selectin-dependent platelet aggregation [22]. In this study, we firmly established that sulfatide binding to P-selectin contributed to platelet aggregation on immobilized collagen under flow and to ex vivo platelet aggregation. Furthermore, we provided an in-depth characterization of the interaction of sulfatides with P-selectin, revealing (1) a critical threshold density of exposed sulfatides required for effective binding to P-selectin, compatible with its focal exposure on the activated platelet membrane and (2) clear differences in sulfatide versus PSGL-1 recognition by P-selectin.

Sulfatides are sulfated glycosphingolipids, composed of a hydrophobic ceramide and a hydrophilic sulfogalactoside unit. Their amphiphilic nature facilitates rapid and quantitative incorporation into liposomes [36]. By multimeric exposure of anionic sulfate moieties, sulfatide liposomes can bind a variety of ligands, including the heparin-binding polypeptide midkine [15], scavenger receptors on macrophages [37], and the blood-brain barrier [38]. We have exploited this feature to characterize the interaction of sulfatides with P-selectin. Interestingly, the liposome sulfatide content, a measure of the exposed anionic surface density, appears to be critical for P-selectin binding. The higher the sulfatide content, the more avid the binding to P- and L-selectin. Analogous to sulfatide binding to midkine (>50–80% sulfatide) [15], a threshold sulfatide density (>21%) exists below which sulfatides do not interact with P- and L-selectin. This finding corresponds to earlier work by our group, in which we showed that multimerization of EWVDV-based peptide ligands enhances their avidity for P-selectin [28]. Conceivably, the assembly of sulfatides into patchy spots upon the platelet surface following activation may be regarded as a dedicated response to meet the threshold density essential for effective interaction with P-selectin. Our findings are in line with observations published by Guchhait et al., who also showed that sulfatides are present in discrete patches on activated spread platelets, possibly like other glycosphingolipids that translocate to membrane microdomains or lipid rafts [39].

The peptide antagonists we used for mapping differences in the P-selectin binding pattern of sulfatides versus PSGL-1, all encompassed the EWVDV core motif critical for P-selectin binding [28]. The smallest fragment to interact with P-selectin and to inhibit PSGL-1 binding in an HL60 cell adhesion assay was EWVDV [28]. Remarkably, while being equally potent in inhibiting PSGL-1-mediated cell adhesion as the non-truncated antagonists, the core motif EWVDV failed to block sulfatide liposome binding to P-selectin and platelet aggregation. As full-length peptides, such as DVEWVDV and CDVEWVDVSC, inhibited both PSGL-1 and sulfatide binding with high potency, the N-terminal extension of EWVDV must be required for inhibition of sulfatide binding as well as platelet aggregation. Intriguingly, CDVEWVDVSC was 20-fold more potent in inhibiting sulfatide than PSGL-1 binding to P-selectin.

The observation that the anti-sulfatide antibody SulphI and the anti-P-selectin antibody AK-4 were able to inhibit platelet aggregation and aggregate size underscored the relevance of the P-selectin/sulfatide dyad for aggregation. P-selectin-dependent inhibition is restricted to the later steps in the aggregation response, suggesting that sulfatides serve a vital role in aggregate stabilization. The observed marked reduction in aggregate size by CDVEWVDVSC and the anti-sulfatide and anti-P-selectin antibodies corresponds with this notion. Interestingly, while equally effective in inhibiting aggregate size, AK-4 appeared to be a much more potent inhibitor of platelet aggregation than SulphI. This could point to an involvement of counterligands other than sulfatides in P-selectin-dependent aggregate formation.

Whereas sulfatide liposome binding could be displaced by antibody WAPS12.2 regardless of sulfatide density, blocking antibody AK-4, EDTA, and CDVEWVDVSC exclusively interfered with P-selectin binding to liposomes containing intermediate sulfatide densities (28%). This is supportive of the existence of at least two separate sulfatide binding domains on P-selectin: a Ca^2+^-dependent binding site in the lectin domain of P-selectin for intermediate-density sulfatide liposomes, and a Ca^2+^-independent binding site for high-density sulfatide liposomes. The peptide antagonists were earlier shown to interact with or in proximity to the sLe^x^ binding site on P-selectin and to inhibit PSGL-1 binding to P-selectin under static and flow conditions [28], and in preliminary work, we were able to pinpoint the peptide binding site to the C-type lectin domain (aa 42–160) of human P-selectin. By analogy with PSGL-1 [40,41], sulfatides may, through their single sulfate group, dock into the well-defined sulfotyrosine binding pocket located at lysine residues 111 and 113, at the C-terminal end of the lectin domain [19,30,42,43,44,45]. This interaction is not targeted by truncated peptide antagonists (e.g., EWVDV), which associate with the sLe^x^ binding pocket [28]. Conversely, the elongated peptide antagonists, such as DVEWVDVS, may protrude more deeply into this sulfotyrosine binding pocket, possibly via interaction of its negatively charged N-terminal aspartic acid with the positively charged lysine residues at position 111 and 113 in P-selectin [19,29] (Figure 6). Epitopes of WAPS12.2 and AK-4 are not known to date, but this information will aid to identify the exact interaction site of sulfatides with P-selectin.

In conclusion, this study established the importance of sulfatides for P-selectin-dependent platelet-platelet interaction. It identified exposed sulfatide density as a key determinant of sulfatide binding and suggested that the interaction of P-selectin with sulfatides, PSGL-1, and sLe^x^ involved multiple binding pockets. Elongated and truncated EWVDV-based peptide antagonists appeared to be selective competitors of sulfatide and PSGL-1 binding, respectively, but only the elongated peptides interdicted P-selectin-mediated platelet aggregation. This renders the elongated EWVDV peptides attractive in therapeutic strategies that aim at reducing thrombus size and stability. Since sulfatides also contribute to platelet-leukocyte tethering [23] and play an important role in various pathophysiological conditions, these peptides will target multiple processes in which activated platelets participate. Finally, the differential activity profile of truncated and elongated peptide antagonists might pave the way to selective intervention in P-selectin-mediated thrombotic responses, leaving P-selectin-driven cell adhesion processes unaffected.

## Figures and Tables

**Figure 1 jcm-08-01266-f001:**
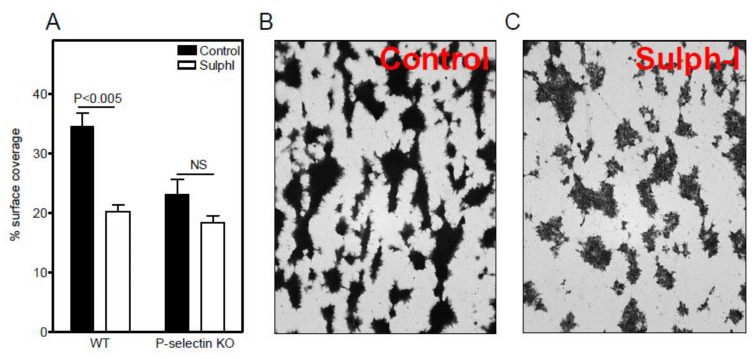
Contribution of sulfatides to P-selectin-dependent aggregate formation of mouse platelets under flow. (**A**) Whole blood from wildtype (WT) and P-selectin knockout (KO) mice was perfused over equine collagen-coated coverslips in a single-passage perfusion chamber in the absence (closed bars) or presence of SulphI antibody (10 µg/mL; open bars) at a shear rate of 150 s^−1^. Coverslips were stained with May-Grünwald-Giemsa and evaluated by light microscopy for % surface coverage. Data are means ± S.E.M. of single perfusion performed in triplicate and are representative for three perfusions. (**B**,**C**) Microscopic high power views showing that WT aggregates were more densely packed in the absence (**B**) than in the presence of SulphI antibody (**C**). NS: not significant.

**Figure 2 jcm-08-01266-f002:**
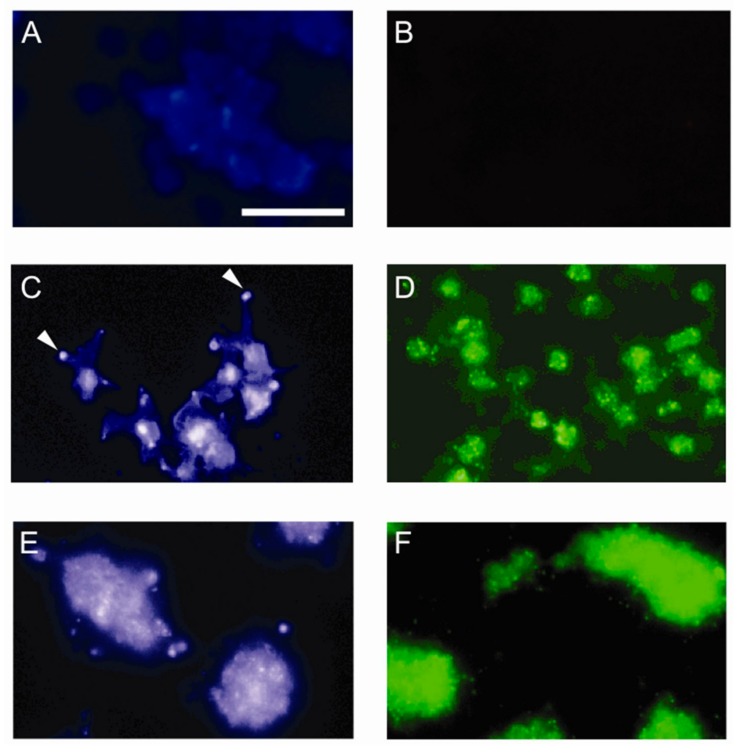
Sulfatide localization. Exposure of cholesterol (panels **A**, **C**, **E**) and sulfatides (panels **B**, **D**, **F**) in human platelets was determined by immunofluorescent labeling using filipin III for cholesterol detection and antibody SulphI for sulfatide detection. The cellular localization in resting human platelets (panels **A** and **B**) was compared with platelets adhered to immobilized fibrinogen (panels **C** and **D**) and collagen (panels **E** and **F**) under flow at a shear rate of 300 s^−1^ and 800 s^−1^, respectively. Arrowheads indicate redistribution of cholesterol in filopodia. (scale bar = 10 μm).

**Figure 3 jcm-08-01266-f003:**
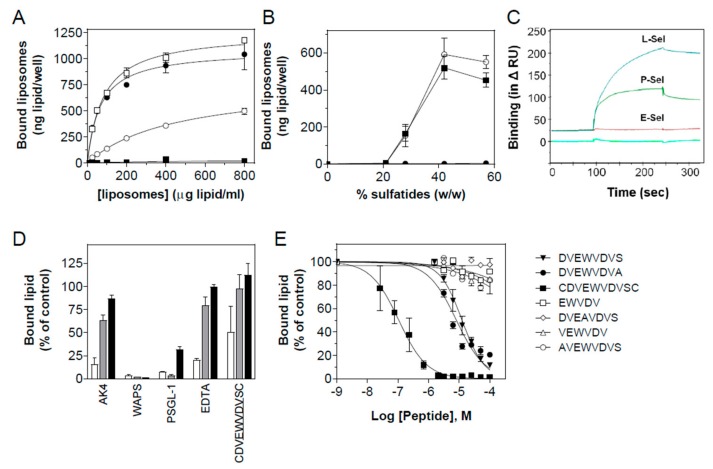
Characterization of sulfatide-P-selectin interaction. (**A**) Binding of sulfatide liposomes to P-selectin-coated wells. [^3^H]-cholesteryl oleate-labeled liposomes were prepared, containing 21 (Sf21;■), 28 (Sf28;○), 42 (Sf42;□), and 57 (Sf57;●) % (*w*/*w*) sulfatides, and were tested for binding to P-selectin-coated wells. (**B**) Binding of sulfatide liposomes to P-, L-, and E-selectin. Sf0, Sf21, Sf28, Sf42, and Sf57 liposomes (~100 µg/mL) were incubated in P-selectin (■), L-selectin (○), or E-selectin (●)-coated microtiter wells. No binding was detected in the control wells. Total bound lipid was calculated from recovered [^3^H]-cholesteryl oleate and its specific activity defined in “Materials and Methods”. Data are means ± S.E.M, n = 3. (**C**) Sulfatide-coated sensor chips bind L- and P-selectin but not E-selectin. Sulfatides were immobilized on an L1 sensor chip at 5 µL/min, and binding kinetics were measured for human Fc fusion proteins of L-selectin (blue), P-selectin (green), E-selectin (red) (all 40 nM). CD154 or CD40 were taken along as negative control but showed no detectable binding. All curves were corrected for nonspecific binding by an online baseline subtraction of ligand binding to a control flow channel. Binding kinetics were analyzed using BIAevaluation software (V2.1; Pharmacia Biosensor, Uppsala, Sweden). (**D**) Interference with sulfatide liposome binding to P-selectin. Binding of Sf28 (white bars), Sf42 (grey bars), and Sf57 (black bars) liposomes (100 μg/mL) to P-selectin-coated microtiter wells was measured in the presence of P-selectin blocking antibodies AK-4 and WAPS12.2 (both 10 μg/mL), rPSGL-1-Ig (15 μg/mL), EDTA (10 mM), or CDVEWVDVSC (0.5 µM). Data are means ± S.E.M, n = 3. (**E**) Competition studies of sulfatide liposome binding to P-selectin. [^3^H]-cholesterol oleate-labeled Sf28 liposomes (1 x 10^5^ dpm) were incubated in P-selectin-coated wells in the absence or presence of increasing amounts of peptides EWVDV (□); VEWVDV (∆); DVEWVDVS (▼); **A**VEWVDVS (○); DVE**A**VDVS (◊); DVEWVDV**A** (●); CDVEWVDVSC (■). Data are expressed as % of total P-selectin binding in the absence of inhibitors (=100%). Data are means ± S.E.M, n = 3.

**Figure 4 jcm-08-01266-f004:**
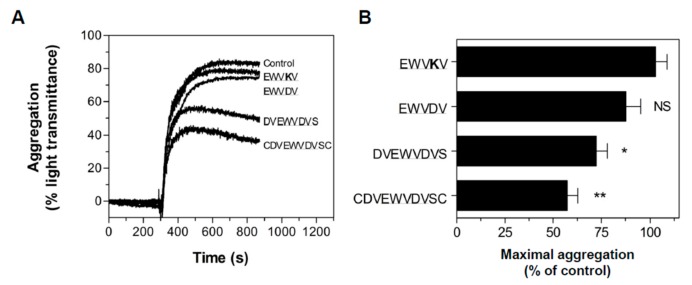
Inhibition of platelet aggregation by P-selectin antagonists. Human platelet-rich plasma was incubated with the indicated peptide antagonists (100 μM) or buffer. Platelet aggregation was initiated by 10 μM adenosine diphosphate (ADP). (**A**) Representative aggregation curves in the presence of buffer, EWV**K**V, EWVDV, DVEWVDVS, and CDVEWVDVSC. (**B**) Maximal aggregation in the presence of EWVDV-based peptides. Data are means ± S.E.M, n = 3; NS: not significant.

**Figure 5 jcm-08-01266-f005:**
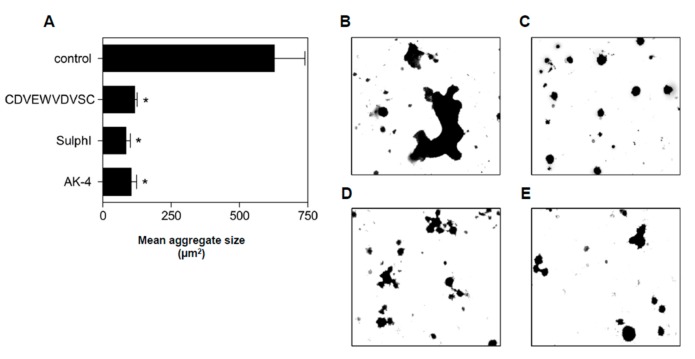
Inhibition of aggregate formation by P-selectin antagonists. (**A**) Mean aggregate size after ADP-induced aggregation of human PRP (platelet-rich plasma) in the absence (control) and presence of CDVEWVDVSC (100 μM), anti-sulfatide antibody SulphI (10 μg/mL), or P-selectin blocking antibody AK-4 (10 μg/mL). Following aggregation, aggregates were applied to microscope slides, stained with May-Grünwald-Giemsa, and analyzed by light microscopy. Mean aggregate size was evaluated by the ‘watershed’ function of the Optimas 6.2 software. Data are means ± S.E.M, n = 3. (**B–E**) Microscopic high power views show aggregate formation in the presence of (**B**) buffer, (**C**) CDVEWVDVSC, (**D**) P-selectin blocking antibody AK-4, and (**E**) anti-sulfatide antibody SulphI.

**Figure 6 jcm-08-01266-f006:**
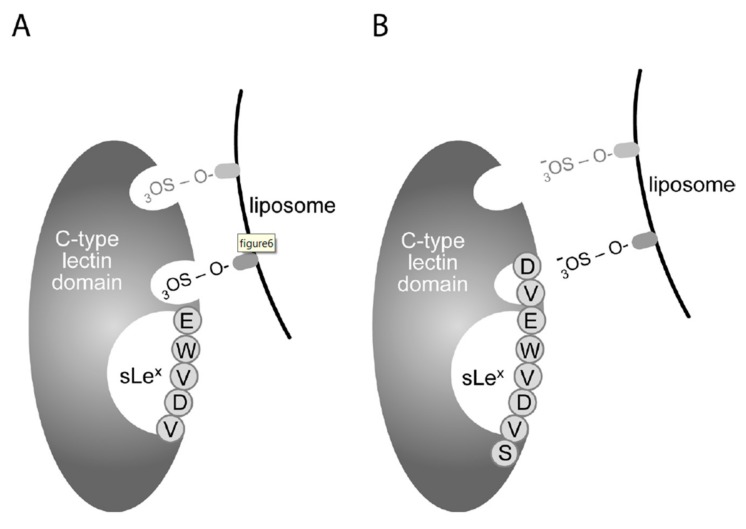
Model of sulfatide binding to P-selectin. A sulfatide moiety in liposome Sf28 docks into one of the defined sulfotyrosine binding sites on P-selectin [40,41,42]. (**A**) Peptide antagonist EWVDV binds to the sLe^x^ binding pocket but does not inhibit the interaction of sulfatides with P-selectin. (**B**) Sulfatide binding is inhibited by elongated peptide antagonist DVEWVDVS, which interacts with the sLe^x^ binding pocket, and may protrude at its N-terminal end into the sulfatide binding pocket. The N-terminal, negatively charged, aspartic acid is probably involved in the interaction with positively charged lysine residues at positions 111 and 113 in P-selectin that mediate interaction with sulfatides [19,28]. The sulfatide moiety indicated in dark grey docks into the Ca^2+^-dependent binding site, while at high-density, a second or even third sulfatide moiety, indicated in light grey, interacts with Ca^2+^-independent binding pockets. The Ca^2+^-dependent interaction of sulfatides with P-selectin is blocked by WAPS12.2, AK-4, and elongated peptides, while the Ca^2+^-independent binding is exclusively blocked by WAPS12.2.

**Table 1 jcm-08-01266-t001:** The ability of EWVDV motif-containing peptides to displace P-selectin binding to HL60 cells (PSGL-1 (P-selectin Glycoprotein Ligand-1)) and sulfatide liposomes (Sf28) and to attenuate platelet aggregation.

Peptide Sequence	PSGL-1 ^*^ EC_50_ (μM)	Sulfatide EC_50_ (μM)	Aggregation ^†^
EWVKV	>1000	>1000	0% (NS ^‡^)
EWVDV	12	>1000	12% (NS)
VEWVDV	13	>1000	Nd ^§^
**A**VEWVDVS	12	>1000	Nd
D**A**EWVDVS	19	>1000	Nd
DVAWVDVS	41	>1000	Nd
DVE AVDVS	>1000	>1000	Nd
DVEWADVS	>1000	>1000	Nd
DVEWVAVS	>1000	>1000	Nd
DVEWVDAS	>1000	>1000	Nd
DVEWVDV**A**	7	8	Nd
DVEWVDVS	11	12	28% (*p*<0.05)
CDVEWVDVSC	9	0.2	43% (*p*<0.001)

^*^ From Molenaar et al. [28] ^†^ Percentage inhibition of platelet aggregation at 100 µM peptide concentration. ^‡^ Not significant. ^§^ Not determined. EC_50_: half maximal effective concentration.

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
