# Peer review of "Peptide Antagonists for P-selectin Discriminate between Sulfatide-Dependent Platelet Aggregation and PSGL-1-Mediated Cell Adhesion"

_jcm, 2019, doi:10.3390/jcm8081266_

Round 1

Reviewer 1 Report

In the present study, authors reported the design of specific peptide antagonists for P-selectin and showed that peptides carrying an EWVDV minimal motif interacted with P-selectin in close proximity to the PSGL-1 binding site on P-selectin, which was proposed to overlap with that of sulfatides. The present study showed that P-selectin binding to sulfatide-laden liposomes requires a threshold sulfatide density, that activated platelets express clusters of surface-exposed sulfatides, and that the binding profile of P-selectin to sulfatides differs from that to PSGL-1. Authors’ experiments showed that elongated but not truncated EWVDV peptide antagonists inhibit sulfatide binding to P-selectin as well as platelet aggregation. However, the following issues remain to be clarified before acceptance.

Major comment:

1.For human platelet experiments, it is described “Freshly drawn venous blood from healthy volunteers was collected with informed consent into 0.1 volume 130 mmol/L trisodium citrate. The donors were free of medication for at least two weeks prior to blood collection.” Was it done with written consent form? Authors need to provide the IRB number.

2.It needs to be described whether human subjects fasted overnight before the blood samples were collected. There are no information on the number of platelets donors in the studyWhat is the interval from time of blood drawing to final test?

3.Authors needs to provide the Animal care number.

4.Does peptide antagonists (e.g. EWVKV, EWVDV, DVEWVDVS, and CDVEWVDVSC) have any cytotoxic effects in platelets?

Reviewer 2 Report

The manuscript "Peptide antagonists for P-selectin discriminate between sulfatide-dependent platelet aggregation and PSGL-1-mediated cell adhesion", written by Korporaal et al. describes a process, where P-selectin-mediated platelet aggregation on a collagen-coated surface depends on present sulfatides. The manuscript is well and carefully written, and the theme is sound and it is worth to pursuit. I have only a few very minor comments:

1. Quality of the figure 3 should be improved. The labels are difficult to read.

2. The figure 3E should be divided into two separate figures to incrase readability. 
